# A GENERALIZATION OF THE RANDOMIZED SINGULAR VALUE DECOMPOSITION

**Nicolas Boullé**
Mathematical Institute
University of Oxford
Oxford, OX2 6GG, UK
`boulle@maths.ox.ac.uk`

**Alex Townsend**
Department of Mathematics
Cornell University
Ithaca, NY 14853, USA
`townsend@cornell.edu`

## ABSTRACT

The randomized singular value decomposition (SVD) is a popular and effective algorithm for computing a near-best rank $k$ approximation of a matrix $A$ using matrix-vector products with standard Gaussian vectors. Here, we generalize the randomized SVD to multivariate Gaussian vectors, allowing one to incorporate prior knowledge of $A$ into the algorithm. This enables us to explore the continuous analogue of the randomized SVD for Hilbert–Schmidt (HS) operators using operator-function products with functions drawn from a Gaussian process (GP). We then construct a new covariance kernel for GPs, based on weighted Jacobi polynomials, which allows us to rapidly sample the GP and control the smoothness of the randomly generated functions. Numerical examples on matrices and HS operators demonstrate the applicability of the algorithm.

## 1 INTRODUCTION

Computing the singular value decomposition (SVD) is a fundamental linear algebra task in machine learning (Paterek, 2007), statistics (Wold et al., 1987), and signal processing (Alter et al., 2000; Van Der Veen et al., 1993). The SVD of an $m \times n$ real matrix $A$ with $m \geq n$ is a factorization of the form $\mathbf{A} = \mathbf{U}\mathbf{\Sigma}\mathbf{V}^*$, where $\mathbf{U}$ is an $m \times m$ orthogonal matrix of left singular vectors, $\mathbf{\Sigma}$ is an $m \times n$ diagonal matrix with entries $\sigma_1(\mathbf{A}) \geq \cdots \geq \sigma_n(\mathbf{A}) \geq 0$, and $\mathbf{V}$ is an $n \times n$ orthogonal matrix of right singular vectors (Golub & Van Loan, 2013). The SVD plays a central role because truncating it after $k$ terms provides a best rank $k$ approximation to $\mathbf{A}$ in the spectral and Frobenius norm (Eckart & Young, 1936; Mirsky, 1960). Since computing the SVD of a large matrix can be computationally infeasible, there are various alternative algorithms that compute near-best rank $k$ matrix approximations from matrix-vector products (Halko et al., 2011; Martinsson & Tropp, 2020; Nakatsukasa, 2020; Nyström, 1930; Williams & Seeger, 2001). The randomized SVD uses matrix-vector products with standard Gaussian random vectors and is one of the most popular algorithms for constructing a low-rank approximation to $A$ (Halko et al., 2011; Martinsson & Tropp, 2020).

Currently, the randomized SVD is used and theoretically justified when it uses matrix-vector products with standard Gaussian random vectors. In this paper, we consider the following generalizations.
**Generalization 1.** We generalize the randomized SVD when the matrix-vector products are with multivariate Gaussian random vectors. Our theory allows for multivariate Gaussian random input vectors that have a general symmetric positive semi-definite covariance matrix. A key novelty of our work is that prior knowledge of $\mathbf{A}$ can be exploited to design covariance matrices that achieve lower approximation errors than the randomized SVD with standard Gaussian vectors.
**Generalization 2.** We generalize the randomized SVD to Hilbert–Schmidt (HS) operators (Hsing & Eubank, 2015). We design a practical algorithm for learning HS operators using random input functions, sampled from a Gaussian process (GP). Examples of applications include learning integral kernels such as Green's functions associated with linear partial differential equations (Boullé & Townsend, 2022; Boullé et al., 2021).

The choice of the covariance kernel in the GP is crucial and impacts both the theoretical bounds and numerical results of the randomized SVD. This leads us to introduce a new covariance kernel

based on weighted Jacobi polynomials for learning HS operators. One of the main advantages of this kernel is that it is directly expressed as a Karhunen–Loève expansion (Karhunen, 1946; Loève, 1946) so that it is faster to sample functions from the associated GP than using a standard squared-exponential kernel. In addition, we show that the smoothness of the functions sampled from a GP with the Jacobi kernel can be controlled as it is related to the decay rate of the kernel's eigenvalues.

**Contributions.** We summarize our novel contributions as follows:

1. We provide new theoretical bounds for the randomized SVD for matrices or HS operators when using random input vectors generated from any multivariate Gaussian distribution. This shows when it is beneficial to use nonstandard Gaussian random vectors in the randomized SVD for constructing low-rank approximations.

2. We generalize the randomized SVD to HS operators and provide numerical examples to learn integral kernels.

3. We propose a covariance kernel based on weighted Jacobi polynomials and show that one can select the smoothness of the sampled random functions by choosing the decay rate of the kernel eigenvalues.

## 2 BACKGROUND: THE RANDOMIZED SVD FOR MATRICES

The randomized SVD computes a near-best rank $k$ approximation to a matrix $A$. First, one performs the matrix-vector products $y_1 = \mathbf{A}x_1, \ldots, y_{k+p} = \mathbf{A}x_{k+p}$, where $x_1, \ldots, x_{k+p}$ are standard Gaussian random vectors with identically and independently distributed entries and $p \geq 1$ is an oversampling parameter. Then, one computes the economized QR factorization $[y_1 \quad \cdots \quad y_{k+p}] = \mathbf{QR}$, before forming the rank $\leq k + p$ approximant $\mathbf{QQ}^*\mathbf{A}$. Note that if $\mathbf{A}$ is symmetric, one can form $\mathbf{QQ}^*\mathbf{A}$ by computing $\mathbf{Q}(\mathbf{AQ})^*$ via matrix-vector products involving $\mathbf{A}$. The quality of the rank $\leq k + p$ approximant $\mathbf{QQ}^*\mathbf{A}$ is characterized by the following theorem.

**Theorem 1 (Halko et al. (2011))** *Let $\mathbf{A}$ be an $m \times n$ matrix, $k \geq 1$ an integer, and choose an oversampling parameter $p \geq 4$. If $\mathbf{\Omega} \in \mathbb{R}^{n \times (k+p)}$ is a standard Gaussian random matrix and $\mathbf{QR} = \mathbf{A\Omega}$ is the economized QR decomposition of $\mathbf{A\Omega}$, then for all $u, t \geq 1$,*

$$\|\mathbf{A} - \mathbf{QQ}^*\mathbf{A}\|_{\mathrm{F}} \leq \left(1 + t\sqrt{\frac{3k}{p+1}}\right)\sqrt{\sum_{j=k+1}^{n} \sigma_j^2(\mathbf{A})} + ut\frac{\sqrt{k+p}}{p+1}\sigma_{k+1}(\mathbf{A}), \qquad (1)$$

*with failure probability at most $2t^{-p} + e^{-u^2}$.*

The squared tail of the singular values of $\mathbf{A}$, i.e., $\sqrt{\sum_{j=k+1}^{n} \sigma_j^2(\mathbf{A})}$, gives the best rank $k$ approximation error to $\mathbf{A}$ in the Frobenius norm. This result shows that the randomized SVD can compute a near-best low-rank approximation to $\mathbf{A}$ with high probability.

## 3 GENERALIZED RANDOMIZED SVD FOR MATRICES AND OPERATORS

The theory behind the randomized SVD has been recently extended to nonstandard covariance matrices and HS operators (Boullé & Townsend, 2022). However, the probability bounds, generalizing Theorem 1, are not sharp enough to emphasize the improved performance of covariance matrices with prior information over the standard randomized SVD. We provide new bounds for GPs with nonstandard covariance matrices in Theorem 2. An upper bound on the expectation is also available in the Appendix (see Proposition 6). While Theorem 2 is formulated with matrices, the same result holds for HS operators in infinite dimensions.

For a fixed target rank $1 \leq k \leq n$, we define $\mathbf{V}_1 \in \mathbb{R}^{n \times k}$ and $\mathbf{V}_2 \in \mathbb{R}^{n \times (n-k)}$ to be the matrices containing the first $k$ and last $n - k$ right singular vectors of $\mathbf{A}$, respectively, and denote by $\mathbf{\Sigma}_2 \in \mathbb{R}^{(n-k) \times (n-k)}$, the diagonal matrix with entries $\sigma_{k+1}(\mathbf{A}), \ldots, \sigma_n(\mathbf{A})$. We consider a symmetric positive semi-definite covariance matrix $\mathbf{K} \in \mathbb{R}^{n \times n}$, with $k$th largest eigenvalue $\lambda_k > 0$.

**Theorem 2** *Let $\mathbf{A}$ be an $m \times n$ matrix, $k \geq 1$ an integer, and choose an oversampling parameter $p \geq 4$. If $\mathbf{\Omega} \in \mathbb{R}^{n \times (k+p)}$ is a Gaussian random matrix, where each column is sampled from a*

*multivariate Gaussian distribution with covariance matrix $\mathbf{K} \in \mathbb{R}^{n \times n}$, and $\mathbf{Q}\mathbf{R} = \mathbf{A}\mathbf{\Omega}$ is the economized QR decomposition of $\mathbf{A}\mathbf{\Omega}$, then for all $u, t \geq 1$,*

$$\|\mathbf{A} - \mathbf{Q}\mathbf{Q}^*\mathbf{A}\|_F \leq \left(1 + ut\sqrt{(k+p)\frac{3k}{p+1}\frac{\beta_k}{\gamma_k}}\right)\sqrt{\sum_{j=k+1}^{n} \sigma_j^2(\mathbf{A})}, \qquad (2)$$

*with failure probability at most $t^{-p} + [ue^{-(u^2-1)/2}]^{k+p}$. Here, $\gamma_k = k/(\lambda_1 \operatorname{Tr}((\mathbf{V}_1^*\mathbf{K}\mathbf{V}_1)^{-1})))$ denotes the covariance quality factor, and $\beta_k = \operatorname{Tr}(\mathbf{\Sigma}_2^2\mathbf{V}_2^*\mathbf{K}\mathbf{V}_2)/(\lambda_1\|\mathbf{\Sigma}_2\|_F^2)$, where $\lambda_1$ is the largest eigenvalue of $\mathbf{K}$.*

The factors $\gamma_k$ and $\beta_k$, measuring the quality of the covariance matrix to learn $\mathbf{A}$ in Theorem 2, can be respectively bounded (Boullé & Townsend 2022, Lem. 2; Lemma 9) using the eigenvalues $\lambda_1 \geq \cdots \geq \lambda_n$ of the covariance matrix $\mathbf{K}$ and the singular values of $\mathbf{A}$ as:

$$\frac{1}{\gamma_k} \leq \frac{1}{k}\sum_{j=n-k+1}^{n} \frac{\lambda_1}{\lambda_j}, \qquad \beta_k \leq \sum_{j=k+1}^{n} \frac{\lambda_{j-k}}{\lambda_1}\sigma_j^2(\mathbf{A}) \Big/ \sum_{j=k+1}^{n} \sigma_j^2(\mathbf{A}).$$

This shows that the performance of the generalized randomized SVD depends on the decay rate of the sequence $\{\lambda_j\}$. The quantities $\gamma_k$ and $\beta_k$ depend on how much prior information of the $k+1, \ldots, n$ right singular vectors of $\mathbf{A}$ is encoded in $\mathbf{K}$. In the ideal situation where these singular vectors are known, then one can define $\mathbf{K}$ such that $\beta_k = 0$ for $\lambda_{k+1} = \cdots = \lambda_n = 0$. In particular, this highlights that a suitably chosen covariance matrix can outperform the randomized SVD with standard Gaussian vectors (see Section 5.1 for a numerical example).

## 3.1 RANDOMIZED SVD FOR HILBERT–SCHMIDT OPERATORS

We now describe the randomized SVD for learning HS operators (see Algorithm 1). The algorithm is implemented in the Chebfun software system (Driscoll et al., 2014), which is a MATLAB package for computing with functions. The Chebfun implementation of the randomized SVD for HS operators uses Chebfun's capabilities, which offer continuous analogues of several matrix operations like the QR decomposition and numerical integration. Indeed, the continuous analogue of a matrix-vector multiplication $\mathbf{A}\mathbf{\Omega}$ for an HS integral operator $\mathscr{F}$ (see Hsing & Eubank, 2015 for definitions and properties of HS operators), with kernel $G : D \times D \to \mathbb{R}$, is

$$(\mathscr{F}f)(x) = \int_D G(x, y)f(y)\,\mathrm{d}y, \quad x \in D, \quad f \in L^2(D),$$

where $D \subset \mathbb{R}^d$ with $d \geq 1$, and $L^2(D)$ is the space of square-integrable functions on $D$.

---

**Algorithm 1** Randomized SVD for HS operators

---

**Input:** HS integral operator $\mathscr{F}$ with kernel $G(x, y)$, number of samples $k > 0$
**Output:** Approximation $G_k$ of $G$
  1: Define a GP covariance kernel $K$
  2: Sample the GP $k$ times to generate a quasimatrix of random functions $\Omega = [f_1 \ldots f_k]$
  3: Evaluate the integral operator at $\Omega$, $Y = [\mathscr{F}(f_1) \ldots \mathscr{F}(f_k)]$
  4: Orthonormalize the columns of $Y$, $Q = \operatorname{orth}(Y) = [q_1 \ldots q_k]$
  5: Compute an approximation to $G$ by evaluating the adjoint of $\mathscr{F}$
  6: Initialize $G_k(x, y)$ to $0$
  7: **for** $i = 1 : k$ **do**
  8:     $G_k(x, y) \leftarrow G_k(x, y) + q_i(x)\int_D G(z, y)q_i(z)\,\mathrm{d}z$
  9: **end for**

---

The algorithm takes as input an integral operator that we aim to approximate. Note that we focus here on learning an integral operator, but other HS operators would work similarly. The first step of the randomized SVD for HS operators consists of generating an $\infty \times k$ quasimatrix $\Omega$ by sampling a GP $k$ times, where $k$ is the target rank (see Section 4). An $\infty \times k$ quasimatrix is an ordered set of $k$ functions (Townsend & Trefethen, 2015), and generalizes the notion of matrix to infinite dimensions. Therefore, each column of $\Omega$ is an object, consisting of a polynomial approximation of

a smooth random function sampled from the GP in the Chebyshev basis. After evaluating the HS operator at $\Omega$ to obtain a quasimatrix $Y$, we use the QR algorithm (Townsend & Trefethen, 2015) to obtain an orthonormal basis $Q$ for the range of the columns of $Y$. Then, the randomized SVD for HS operators requires the left-evaluation of the operator $\mathscr{F}$ or, equivalently, the evaluation of its adjoint $\mathscr{F}_t$ satisfying:

$$(\mathscr{F}_t f)(x) = \int_D G(y, x) f(y) \, \mathrm{d}y, \quad x \in D.$$

We evaluate the adjoint of $\mathscr{F}$ at each column vector of $Q$ to construct an approximation $G_k$ of $G$. Finally, the approximation error between the operator kernel $G$ and the learned kernel $G_k$ can be computed in the $L^2$-norm, corresponding to the HS norm of the integral operator.

## 4    COVARIANCE KERNELS

To generate the random input functions $f_1, \ldots, f_k$ for the randomized SVD for HS operators, we draw them from a GP, denoted by $\mathcal{GP}(0, K)$, for a certain covariance kernel $K$. A widely employed covariance kernel is the squared-exponential function $K_{\mathrm{SE}}$ (Rasmussen & Williams, 2006) given by

$$K_{\mathrm{SE}}(x, y) = \exp\left(-|x - y|^2/(2\ell^2)\right), \quad x, y \in D, \tag{3}$$

where $\ell > 0$ is a parameter controlling the length-scale of the GP. This kernel is isotropic as it only depends on $|x - y|$, is infinitely differentiable, and its eigenvalues decay supergeometrically to 0. Since the bound in Theorem 2 degrades as the ratio $\lambda_1/\lambda_j$ increases for $j \geq k + 1$, the randomized SVD for learning HS operators prefers covariance kernels with slowly decaying eigenvalues. Our randomized SVD cannot hope to learn HS operators where the range of the operator has a rank greater than $\tilde{k}$, where $\tilde{k}$ is such that the $\tilde{k}$th eigenvalue of $K_{\mathrm{SE}}$ reaches machine precision.

Other popular kernels for GPs include the Matérn kernel (Rasmussen & Williams, 2006, Chapt. 4) and Brownian bridge (Nelsen & Stuart, 2021). Prior information on the HS operator can also be enforced through the choice of the covariance kernel. For instance, one can impose the periodicity of the samples by using the following squared-exponential periodic kernel:

$$K_{\mathrm{Per}}(x, y) = \exp\left(-\frac{2}{\ell^2} \sin^2\left(\frac{x - y}{2}\right)\right), \quad x, y \in D,$$

where $\ell > 0$ is the length-scale parameter.

### 4.1    SAMPLE RANDOM FUNCTIONS FROM A GAUSSIAN PROCESS

In finite dimensions, a random vector $u \sim \mathcal{N}(0, \mathbf{K})$, where $\mathbf{K} \in \mathbb{R}^{n \times n}$ is a covariance matrix with Cholesky factorization $\mathbf{K} = \mathbf{L}\mathbf{L}^*$, can be generated from the matrix-vector product $u = \mathbf{L}c$. Here, $c \in \mathbb{R}^n$ is a vector whose entries follow the standard Gaussian distribution. We now detail how this process extends to infinite dimensions with a continuous covariance kernel. Let $K$ be a continuous symmetric positive-definite covariance function defined on the domain $[a, b] \times [a, b] \subset \mathbb{R}^2$ with $-\infty < a < b < \infty$. We consider the continuous analogue of the Cholesky factorization to write $K$ as (Townsend & Trefethen, 2015)

$$K(x, y) = \sum_{j=1}^{\infty} r_j(x) r_j(y) = L_c(x) L_c^*(y), \quad x, y \in [a, b],$$

where $r_j$ is the $j$th row of $L_c$, which —in Chebfun's terminology— is a lower-triangular quasimatrix. In practice, we truncate the series after $n$ terms, either arbitrarily or when the $n$th largest kernel eigenvalue, $\lambda_n$, falls below machine precision. Then, if $c \in \mathbb{R}^n$ follows the standard Gaussian distribution, a function $u$ can be sampled from $\mathcal{GP}(0, K)$ as $u = L_c c$. That is,

$$u(x) = \sum_{j=1}^{n} c_j r_j(x), \quad x \in [a, b].$$

The continuous Cholesky factorization is implemented in Chebfun2 (Townsend & Trefethen, 2013), which is the extension of Chebfun for computing with two-dimensional functions. As an example,

the polynomial approximation, which is accurate up to essentially machine precision, of the squared-exponential covariance kernel $K_{SE}$ with parameter $\ell = 0.01$ on $[-1, 1]^2$ yields a numerical rank of $n = 503$. The functions sampled from $\mathcal{GP}(0, K_{SE})$ become more oscillatory as the length-scale parameter $\ell$ decreases and hence the numerical rank of the kernel increases or, equivalently, the associated eigenvalues sequence $\{\lambda_j\}$ decays slower to zero.

## 4.2 INFLUENCE OF THE KERNEL'S EIGENVALUES AND MERCER'S REPRESENTATION

The covariance kernel can also be defined from its Mercer's representation as

$$K(x, y) = \sum_{j=1}^{\infty} \lambda_j \psi_j(x) \psi_j(y), \quad x, y \in D, \tag{4}$$

where $\{\psi_j\}$ is an orthonormal basis of $L^2(D)$ and $\lambda_1 \geq \lambda_2 \geq \cdots > 0$ (Hsing & Eubank, 2015, Thm. 4.6.5). We prefer to construct $K$ directly from Mercer's representations for several reasons: 1. One can impose prior knowledge of the kernel of the HS operator on the eigenfunctions of $K$ (such as periodicity or smoothness), 2. One can often generate samples from $\mathcal{GP}(0, K)$ efficiently using Equation (4), and 3. One can control the decay rate of the eigenvalues of $K$,

Hence, the quantity $\gamma_k$ in the probability bound of Theorem 2 measures the quality of the covariance kernel $K$ in $\mathcal{GP}(0, K)$ to generate random functions that can learn the HS operator $\mathcal{F}$. To minimize $1/\gamma_k$ we would like to select the eigenvalues $\lambda_1 \geq \lambda_2 \geq \cdots > 0$ of $K$ so that they have the slowest possible decay rate while maintaining $\sum_{j=1}^{\infty} \lambda_j < \infty$. One needs $\{\lambda_j\} \in \ell^1$ to guarantee that $\omega \sim \mathcal{GP}(0, K)$ has finite expected squared $L^2$-norm, i.e., $\mathbb{E}[\|\omega\|_{L^2(D)}^2] = \sum_{j=1}^{\infty} \lambda_j < \infty$. The best sequence of eigenvalues we know that satisfies this property is called the Rissanen sequence (Rissanen, 1983) and is given by $\lambda_j = R_j := 2^{-L(j)}$, where

$$L(j) = \log_2(c_0) + \log_2^*(j), \quad \log_2^*(j) = \sum_{i=2}^{\infty} \max(\log_2^{(i)}(j), 0), \quad c_0 = \sum_{i=2}^{\infty} 2^{-\log_2^*(i)},$$

and $\log_2^{(i)}(j) = \log_2 \circ \cdots \circ \log_2(j)$ is the composition of $\log_2(\cdot)$ $i$ times. Other options for the choice of eigenvalues include any sequence of the form $\lambda_j = j^{-\nu}$ for $\nu > 1$.

## 4.3 JACOBI COVARIANCE KERNEL

If $D = [-1, 1]$, then a natural choice of orthonormal basis of $L^2(D)$ to define the Mercer's representation of the kernel are weighted Jacobi polynomials (Deheuvels & Martynov, 2008; Olver et al., 2010). That is, for a weight function $w_{\alpha,\beta}(x) = (1-x)^\alpha (1+x)^\beta$ with $\alpha, \beta > -1$, and any positive eigenvalue sequence $\{\lambda_j\}$, we consider the Jacobi kernel

$$K_{Jac}^{(\alpha,\beta)}(x, y) = \sum_{j=0}^{\infty} \lambda_{j+1} w_{\alpha,\beta}^{1/2}(x) \tilde{P}_j^{(\alpha,\beta)}(x) w_{\alpha,\beta}^{1/2}(y) \tilde{P}_j^{(\alpha,\beta)}(y), \quad x, y \in [-1, 1], \tag{5}$$

where $\tilde{P}_j^{(\alpha,\beta)}$ is the scaled Jacobi polynomial of degree $j$. The polynomials are normalized such that $\|w_{\alpha,\beta}^{1/2} \tilde{P}_j^{(\alpha,\beta)}\|_{L^2([-1,1])} = 1$ and $K_{Jac}^{(\alpha,\beta)} \in L^2([-1,1]^2)$. In this case, a random function can be sampled as

$$u(x) = \sum_{j=0}^{\infty} \sqrt{\lambda_{j+1}} c_j w_{\alpha,\beta}^{1/2}(x) \tilde{P}_j^{(\alpha,\beta)}(x), \quad x \in [-1, 1],$$

where $c_j \sim \mathcal{N}(0, 1)$ for $0 \leq j \leq \infty$.

A desirable property of a covariance kernel is to be unbiased towards one spatial direction, i.e., $K(x, y) = K(-y, -x)$ for $x, y \in [-1, 1]$, which motivates us to always select $\alpha = \beta$. Moreover, it is desirable to have the eigenfunctions of $K_{Jac}^{(\alpha,\beta)}$ to be polynomial so that one can generate samples from $\mathcal{GP}(0, K)$ efficiently. This leads us to choose $\alpha$ and $\beta$ to be even integers. The choice of $\alpha = \beta = 0$ gives the Legendre kernel (Foster et al., 2020; Habermann, 2021). In Section 5, we use Equation (5) with $\alpha = \beta = 2$ (see Figure 1) to ensure that functions sampled from the associated GP satisfy homogeneous Dirichlet boundary conditions. In this case, one must have $\sum_{j=1}^{\infty} j\lambda_j < \infty$ so that the series of functions in Equation (5) converges uniformly and $K_{Jac}^{(2,2)}$ is a continuous kernel

(see Appendix B). Under this additional constraint, the best choice of eigenvalues is given by a scaled Rissanen sequence: $\lambda_j = R_j/j$, for $j \geq 1$ (cf. Section 4.2). Covariance kernels on higher dimensional domains of the form $D = [-1, 1]^d$, for $d \geq 2$, can be defined using tensor products of weighted Jacobi polynomials.

## 4.4 SMOOTHNESS OF FUNCTIONS SAMPLED FROM A GP WITH JACOBI KERNEL

We now connect the decay rate of the eigenvalues of the Jacobi covariance kernel $K_{\text{Jac}}^{(2,2)}$ to the smoothness of the samples from $\mathcal{GP}(0, K_{\text{Jac}}^{(2,2)})$. Hence, the Jacobi covariance function allows the control of the decay rate of the eigenvalues $\{\lambda_j\}$ as well as the smoothness of the resulting randomly generated functions. First, Lemma 3 asserts that if the coefficients of an infinite polynomial series have sufficient decay, then the resulting series is smooth with regularity depending on the decay rate. This result can be seen as a reciprocal to (Trefethen, 2019, Thm. 7.1).

**Lemma 3** *Let $\{p_j\}$ be a family of polynomials such that $\deg(p_j) \leq j$ and $\max_{x \in [-1,1]} |p_j(x)| = 1$. If $f_n(x) = \sum_{j=0}^{n} a_j p_j(x)$ with $|a_j| \leq j^{-\nu}$ for $\nu > 1$, then $f_n$ converges uniformly to $f(x) = \sum_{j=0}^{\infty} a_j p_j(x)$ and $f$ is $\mu$ times continuously differentiable for any integer $\mu$ such that $\mu < (\nu-1)/2$.*

Note that the main application of this lemma occurs when $\deg(p_j) = j$ for all $j \geq 0$. The proof of Lemma 3 is deferred to the supplementary material. We now state the following theorem about the regularity of functions sampled from $\mathcal{GP}(0, K_{\text{Jac}}^{(2,2)})$, which guarantees that if the eigenvalues are chosen such that $\lambda_j = \mathcal{O}(1/j^\nu)$ with $\nu > 3$, then $f \sim \mathcal{GP}(0, K_{\text{Jac}}^{(2,2)})$ is almost surely continuous. Moreover, a faster decay of the eigenvalues of $K_{\text{Jac}}^{(2,2)}$ implies higher regularity of the sampled functions, in an almost sure sense.

**Theorem 4** *Let $\{\lambda_j\} \in \ell^1(\mathbb{R}^+)$ be a positive sequence such that $\lambda_j = \mathcal{O}(j^{-\nu})$ for $\nu > 3$. If $f$ is sampled from $\mathcal{GP}(0, K_{Jac}^{(2,2)})$, then $f \in \mathcal{C}^\mu([-1, 1])$ almost surely for any integer $\mu < (\nu - 3)/2$.*

This theorem can be seen as a particular case of Driscoll's zero-one law (Driscoll, 1973), which characterizes the regularity of functions samples from GPs (see also Kanagawa et al., 2018).

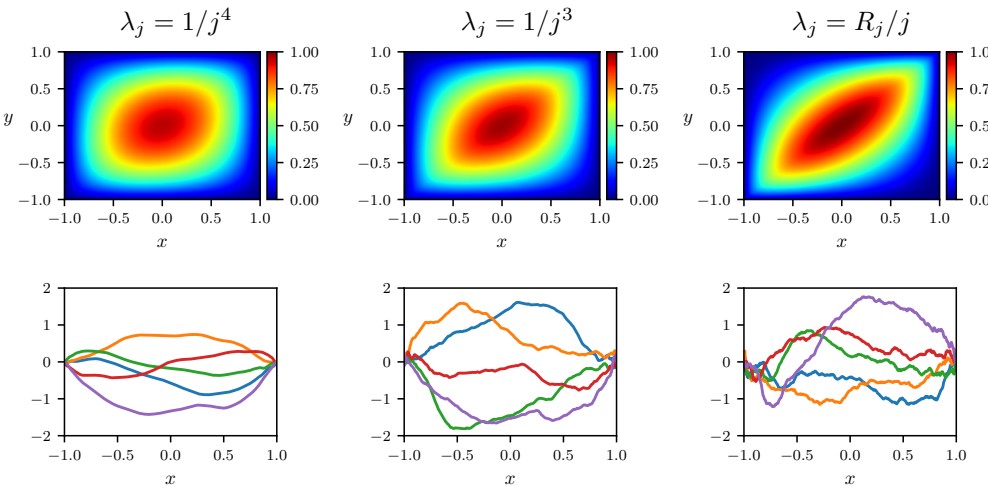

Figure 1: Covariance kernel $K_{\text{Jac}}^{(2,2)}$ constructed using Jacobi polynomials of type $(2, 2)$ with $\lambda_j = 1/j^4$, $1/j^3$, and $R_j/j$, where $R_j$ is the Rissanen sequence (top). The bottom panels illustrate functions sampled from $\mathcal{GP}(0, K_{\text{Jac}}^{(2,2)})$ with the different eigenvalue sequences. The series for generating the random functions are truncated to $n = 500$.

In Figure 1, we display the Jacobi kernel of type $(2, 2)$ with functions sampled from the corresponding GP. We selected eigenvalue sequences of different decay rates: from the faster $1/j^4$ to the slower Rissanen sequence $R_j/j$ (Section 4.2). For $\lambda_j = 1/j^3$ and $\lambda_j = R_j/j$, we observe a large variation of the randomly generated functions near $x = \pm 1$, indicating a potential discontinuity of the samples at these two points as $n \to \infty$. This is in agreement with Theorem 4, which only guarantees continuity (with probability one) of the randomly generated functions if $\lambda_j \sim 1/j^\nu$ with $\nu > 3$.

## 5 NUMERICAL EXPERIMENTS

### 5.1 APPROXIMATION OF MATRICES USING NON-STANDARD COVARIANCE FUNCTIONS

The approximation error bound in Theorem 2 depends on the eigenvalues of the covariance matrix, which dictates the distribution of the column vectors of the input matrix $\mathbf{\Omega}$. Roughly speaking, the more prior knowledge of the matrix $\mathbf{A}$ that can be incorporated into the covariance matrix, the better. In this numerical example, we investigate whether the standard randomized SVD, which uses the identity as its covariance matrix, can be improved by using a different covariance matrix. We then attempt to learn the discretized $2000 \times 2000$ matrix, i.e., the discrete Green's function, of the inverse of the following differential operator:

$$\mathcal{L}u = d^2 u/dx^2 - 100\sin(5\pi x)u, \quad x \in [0,1].$$

We vary the number of columns (i.e. samples from the GP) in the input matrix $\mathbf{\Omega}$ from 1 to 2000.

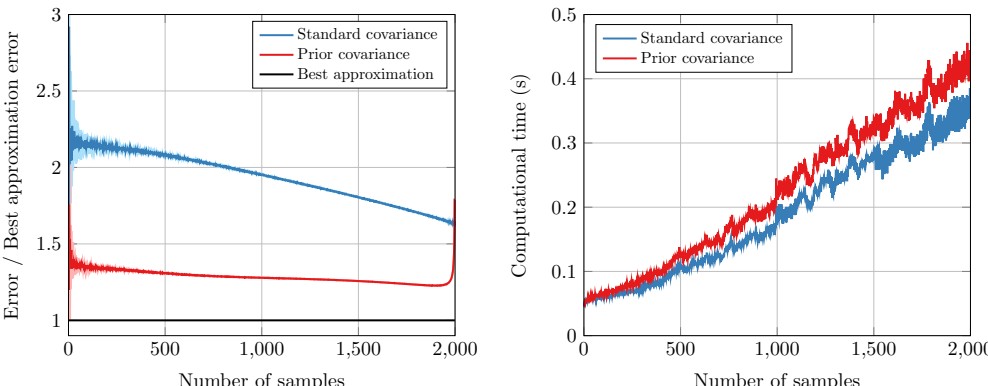

Figure 2: Left: Ratio between the average randomized SVD approximation error (over 10 runs) of the $2000 \times 2000$ matrix of the inverse of the differential operator $\mathcal{L}u = d^2 u/dx^2 - 100\sin(5\pi x)u$ on $[0,1]$, and the best approximation error. The error bars in light colour (blue and red) illustrate one standard deviation. Right: Average computational time of the algorithm (over 10 runs). The eigenvalue decomposition of the covariance matrix has been precomputed before.

In Figure 2(left), we compare the ratios between the relative error in the Frobenius norm given by the randomized SVD and the best approximation error, obtained by truncating the SVD of $\mathbf{A}$. The prior covariance matrix $\mathbf{K}$ consists of the discretized $2000 \times 2000$ matrix of the Green's function of the negative Laplace operator $\mathcal{L}u = -d^2 u/dx^2$ on $[0,1]$ to incorporate knowledge of the diffusion term in the matrix $\mathbf{A}$. We see that a nonstandard covariance matrix leads to a higher approximation accuracy, with a reduction of the error by a factor of 1.3-1.6 compared to the standard randomized SVD. At the same time, the procedure is only 20% slower[1] on average (Figure 2 (right)) as one can precompute the eigenvalue decomposition of the covariance matrix (see Appendix C). It is of interest to maximize the accuracy of the approximation matrix from a limited number of samples in applications where the sampling time is much higher than the numerical linear algebra costs.

### 5.2 APPLICATIONS OF THE RANDOMIZED SVD FOR HILBERT–SCHMIDT OPERATORS

We now apply the randomized SVD for HS operators to learn kernels of integral operators. In this first example, the kernel is defined as (Townsend, 2013)

$$G(x,y) = \cos(10(x^2 + y))\sin(10(x + y^2)), \quad x, y \in [-1, 1],$$

and is displayed in Figure 3(a). We employ the squared-exponential covariance kernel $K_{\mathrm{SE}}$ with parameter $\ell = 0.01$ and $k = 100$ samples (see Equation (3)) to sample random functions from the associated GP. The learned kernel $G_k$ is represented on the bottom panel of Figure 3(a) and has an approximation error around machine precision.

---

[1]Timings were performed on an Intel Xeon CPU E5-2667 v2 @ 3.30GHz using MATLAB R2020b without explicit parallelization.

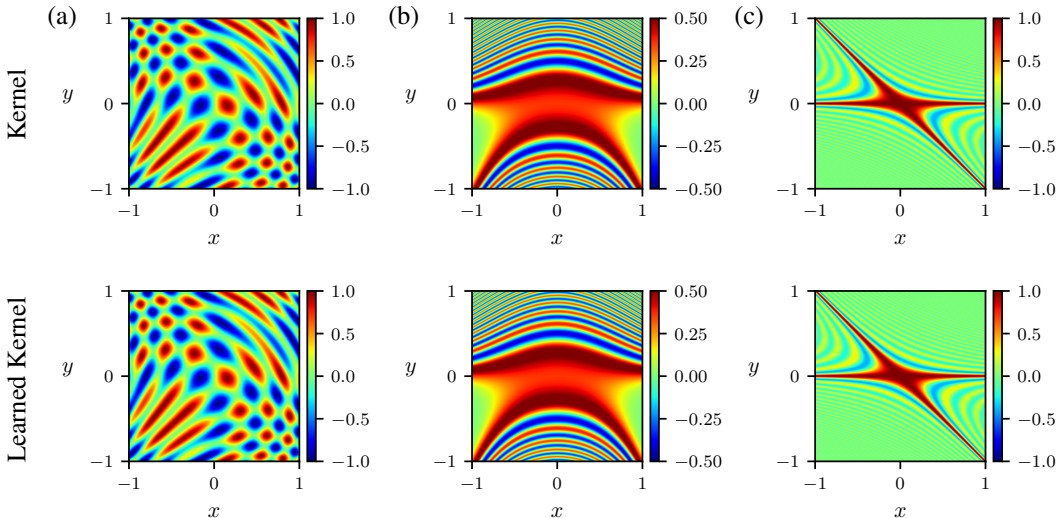

Figure 3: Kernels of three HS operators (top) together with the kernels learned by the randomized SVD for HS operators (bottom), using the squared-exponential covariance kernel $K_{\text{SE}}$ with parameter $\ell = 0.01$ and one hundred functions sampled from $\mathcal{GP}(0, K_{\text{SE}})$.

As a second application of the randomized SVD for HS operators, we learn the kernel $G(x, y) = \text{Ai}(-13(x^2 y + y^2))$ for $x, y \in [-1, 1]$, where Ai is the Airy function (NIS, Chapt. 9) defined by

$$\text{Ai}(x) = \frac{1}{\pi} \int_0^\infty \cos\left(\frac{t^3}{3} + xt\right) \, \mathrm{d}t, \quad x \in \mathbb{R}.$$

We plot the kernel and its low-rank approximant given by the randomized SVD for HS operators in Figure 3(b) and obtain an approximation error (measured in the $L^2$-norm) of $5.04 \times 10^{-14}$. The two kernels have a numerical rank equal to $42$.

The last example consists of learning the HS operator associated with the kernel $G(x, y) = J_0(100(xy + y^2))$ for $x, y \in [-1, 1]$, where $J_0$ is the Bessel function of the first kind (NIS, Chapt. 10) defined as

$$J_0(x) = \frac{1}{\pi} \int_0^\pi \cos(x \sin t) \, \mathrm{d}t, \quad x \in \mathbb{R},$$

and plotted in Figure 3(c). The rank of this kernel is equal to $91$ while its approximation is of rank $89$ and the approximation error is equal to $4.88 \times 10^{-13}$. We observe that in the three numerical examples displayed in Figure 3, the difference between the learned and the original kernel is visually not perceptible.

Finally, we evaluate the influence of the choice of covariance kernel and number of samples in Figure 4. Here, we vary the number of samples from $k = 1$ to $k = 100$ and use the randomized SVD for HS operators with four different covariance kernels: the squared-exponential $K_{\text{SE}}$ with parameters $\ell = 0.01, 0.1, 1$, and the Jacobi kernel $K_{\text{Jac}}^{(2,2)}$ with eigenvalues $\lambda_j = 1/j^3$, for $j \geq 1$. In the left panel of Figure 4, we represent the eigenvalue ratio $\lambda_j / \lambda_1$ of the four kernels and observe that this quantity falls below machine precision for the squared-exponential kernel with $\ell = 1$ and $\ell = 0.1$ at $j = 13$ and $j = 59$, respectively. In Figure 4 (right), we observe that these two kernels fail to approximate kernels of high numerical rank. The other two kernels have a much slower decay of eigenvalues and can capture (or learn) more complicated kernels. We then see in the right panel of Figure 4 that the relative approximation errors obtained using $K_{\text{Jac}}^{(2,2)}$ and $K_{\text{SE}}$ are close to the best approximation error given by the squared tail of the singular values of the integral kernel $G(x, y)$, i.e., $(\sum_{j \geq k+1} \sigma_j^2)^{1/2}$. The overshoot in the error at $k = 100$ compared to the machine precision is due to the decay of the eigenvalues of the covariance kernels. Hence, spatial directions associated with small eigenvalues are harder to learn accurately. This issue does not arise in finite dimensions with the standard randomized SVD because the covariance kernel used there

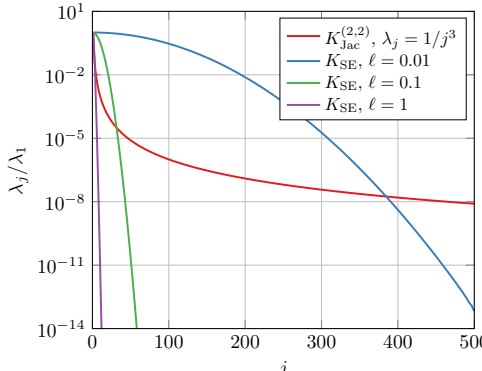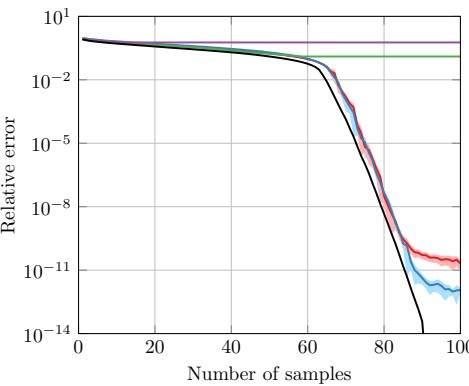

Figure 4: Left: Scaled eigenvalues of the Jacobi covariance kernel $K_{\text{Jac}}^{(2,2)}$ with sequence $\lambda_j = 1/j^3$ and squared-exponential kernels $K_{\text{SE}}$ with parameters $\ell = 0.01, 0.1, 1$, respectively. Right: Average (over 10 runs) relative approximation error in the $L^2$-norm between the Bessel kernel $G(x, y) = J_0(100(xy + y^2))$ and its low-rank approximation $G_k(x, y)$, obtained from the randomized SVD by sampling the GPs $k$ times. The error bars in light colour (blue and red) illustrate one standard deviation and the black line indicates the best approximation error given by the tail of the singular values of $G$.

is isotropic, i.e., all its eigenvalues are equal to one. However, this choice is no longer possible for learning HS integral operators as the covariance kernel $K$ must be squared-integrable. The relative approximation errors at $k = 100$ (averaged over 10 runs) using $K_{\text{Jac}}^{(2,2)}$ and $K_{\text{SE}}$ with $\ell = 0.01$ are $\text{Error}(K_{\text{Jac}}^{(2,2)}) \approx 2.6 \times 10^{-11}$, and $\text{Error}(K_{\text{SE}}) \approx 5.7 \times 10^{-13}$, which gives a ratio of

$$\text{Error}(K_{\text{Jac}}^{(2,2)})/\text{Error}(K_{\text{SE}}) \approx 45.6. \tag{6}$$

However, the square-root of the ratio of the quality of the two kernels for $k = 91$ is equal to

$$\sqrt{\gamma_{91}(K_{\text{SE}})/\gamma_{91}(K_{\text{Jac}}^{(2,2)})} \approx 117.8, \tag{7}$$

which is of the same order of magnitude of Equation (6) as predicted by Theorem 2. In Equation (7), $\gamma_{91}(K_{\text{SE}}) \approx 5.88 \times 10^{-2}$ and $\gamma_{91}(K_{\text{Jac}}^{(2,2)}) \approx 4.24 \times 10^{-6}$ are both computed using Chebfun.

In conclusion, this section provides numerical insights to motivate the choice of the covariance kernel to learn HS operators. Following Figure 4, a kernel with slowly decaying eigenvalues is preferable and yields better approximation errors or higher learning rate with respect to the number of samples, especially when learning a kernel with a large numerical rank. The optimal choice from a theoretical viewpoint is to select a covariance kernel whose eigenvalues have a decay rate similar to the Rissanen sequence (Rissanen, 1983), but other choices may be preferable in practice to ensure smoothness of the sample functions (cf. Section 4.4).

## 6 CONCLUSIONS

We have explored practical extensions of the randomized SVD to nonstandard covariance matrices and Hilbert–Schmidt operators. This paper motivates new computational and algorithmic approaches for preconditioning the covariance kernel based on prior information to compute a low-rank approximation of matrices and impose properties on the learned matrix and random functions from the GP. Numerical experiments demonstrate that covariance matrices with prior knowledge outperform the standard identity matrix used in the literature and lead to near-optimal approximation errors. In addition, we proposed a covariance kernel based on weighted Jacobi polynomials, which allows the control of the smoothness of the random functions generated and may find practical applications in PDE learning (Boullé et al., 2020; 2021) as it imposes prior knowledge of Dirichlet boundary conditions. The algorithm presented in this paper is limited to matrices and HS operators and does not extend to unbounded operators such as differential operators. Additionally, the theoretical bounds only offer probabilistic guarantees for Gaussian inputs, while sub-Gaussian distributions (Kahane, 1960) of the inputs would be closer to realistic application settings.

ACKNOWLEDGMENTS

We thank Joel Tropp for his suggestions leading to Lemma 7 and Daniel Kressner for his comments. This work was supported by the EPSRC Centre For Doctoral Training in Industrially Focused Mathematical Modelling (EP/L015803/1) in collaboration with Simula Research Laboratory, and the National Science Foundation grants DMS-1818757, DMS-1952757, and DMS-2045646.

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

## A    RANDOMIZED SVD WITH MULTIVARIATE GAUSSIAN INPUTS

Let $m \geq n \geq 1$ and $\mathbf{A}$ be an $m \times n$ real matrix with singular value decomposition $\mathbf{A} = \mathbf{U\Sigma V}^*$, where $\mathbf{U}$ and $\mathbf{V}$ are orthonormal matrices, and $\mathbf{\Sigma}$ be an $m \times n$ diagonal matrix with entries $\sigma_1(\mathbf{A}) \geq \cdots \geq \sigma_n(\mathbf{A}) > 0$. For a fixed target rank $k \geq 1$, we define $\mathbf{\Sigma}_1$ and $\mathbf{\Sigma}_2$ to be the diagonal matrices, which respectively contain the first $k$ singular values of $\mathbf{A}$: $\sigma_1(\mathbf{A}) \geq \cdots \geq \sigma_k(\mathbf{A})$, and the remaining singular values. Let $\mathbf{V}_1$ be the $n \times k$ matrix obtained by truncating $\mathbf{V}_1$ after $k$ columns and $\mathbf{V}_2$ the remainder. In this section, $\mathbf{K}$ denotes a symmetric positive semi-definite $n \times n$ matrix and $\mathbf{\Omega} \in \mathbb{R}^{n \times \ell}$ a Gaussian random matrix with $\ell \geq k$ independent columns sampled from a multivariate normal distribution with covariance matrix $\mathbf{K}$. Finally, we define $\mathbf{\Omega}_1 \coloneqq \mathbf{V}_1^*\mathbf{\Omega}$ and $\mathbf{\Omega}_2 \coloneqq \mathbf{V}_2^*\mathbf{\Omega}$. We first refine (Boullé & Townsend, 2022, Lem. 5).

**Lemma 5** *Let $\ell \geq 1$, $\mathbf{\Omega} \in \mathbb{R}^{n \times \ell}$ be a Gaussian random matrix, where each column is sampled from a multivariate Gaussian distribution with covariance matrix $\mathbf{K}$, and $\mathbf{T}$ be an $\ell \times k$ matrix. Then,*

$$\mathbb{E}[\|\mathbf{\Sigma}_2\mathbf{V}_2^*\mathbf{\Omega T}\|_{\mathrm{F}}^2] = \mathrm{Tr}(\mathbf{\Sigma}_2^2\mathbf{V}_2^*\mathbf{KV}_2)\|\mathbf{T}\|_{\mathrm{F}}^2. \tag{8}$$

**Proof.** Let $\mathbf{K} = \mathbf{Q_K}\mathbf{\Lambda}\mathbf{Q_K^*}$ be the eigenvalue decomposition of $\mathbf{K}$, where $\mathbf{Q_K}$ is orthonormal and $\mathbf{\Lambda}$ is a diagonal matrix containing the eigenvalues of $\mathbf{K}$ in decreasing order: $\lambda_1 \geq \cdots \geq \lambda_n \geq 0$. We note that $\mathbf{\Omega}$ can be expressed as $\mathbf{\Omega} = \mathbf{Q_K}\mathbf{\Lambda}^{1/2}\mathbf{G}$, where $\mathbf{G}$ is a standard Gaussian matrix. Let $\mathbf{S} = \mathbf{\Sigma}_2\mathbf{V}_2^*\mathbf{Q_K}\mathbf{\Lambda}^{1/2}$, the proof follows from (Halko et al., 2011, Prop. A.1), which shows that $\mathbb{E}\|\mathbf{SGT}\|_{\mathrm{F}}^2 = \|\mathbf{S}\|_{\mathrm{F}}^2\|\mathbf{T}\|_{\mathrm{F}}^2$. ∎

Note that one can bound the term $\mathrm{Tr}(\mathbf{\Sigma}_2^2\mathbf{V}_2^*\mathbf{KV}_2)$ by $\lambda_1\|\mathbf{\Sigma}_2\|_{\mathrm{F}}^2$, where $\lambda_1$ is the largest eigenvalue of $\mathbf{K}$ (Boullé & Townsend, 2022). While this provides a simple upper bound, it does not demonstrate that the use of a covariance matrix containing prior information on the singular vectors of $\mathbf{A}$ can outperform the randomized SVD with standard Gaussian inputs. Then, combining the proof of (Boullé & Townsend, 2022, Thm. 1) and Lemma 5, we prove the following proposition, which bounds the expected approximation error of the randomized SVD with multivariate Gaussian inputs.

**Proposition 6** *Let $\mathbf{A}$ be an $m \times n$ matrix, $k \geq 1$ an integer, and choose an oversampling parameter $p \geq 2$. If $\mathbf{\Omega} \in \mathbb{R}^{n \times (k+p)}$ is a Gaussian random matrix, where each column is sampled from a multivariate Gaussian distribution with covariance matrix $\mathbf{K} \in \mathbb{R}^{n \times n}$, and $\mathbf{QR} = \mathbf{A\Omega}$ is the economized QR decomposition of $\mathbf{A\Omega}$, then,*

$$\mathbb{E}\left[\|\mathbf{A} - \mathbf{QQ}^*\mathbf{A}\|_{\mathrm{F}}\right] \leq \left(1 + \sqrt{\frac{\beta_k}{\gamma_k}\frac{k(k+p)}{p-1}}\right)\sqrt{\sum_{j=k+1}^{n}\sigma_j^2(\mathbf{A})},$$

*where $\gamma_k = k/(\lambda_1\,\mathrm{Tr}((\mathbf{V}_1^*\mathbf{KV}_1)^{-1}))$ and $\beta_k = \mathrm{Tr}(\mathbf{\Sigma}_2^2\mathbf{V}_2^*\mathbf{KV}_2)/(\lambda_1\|\mathbf{\Sigma}_2\|_{\mathrm{F}}^2)$.*

We remark that for standard Gaussian inputs, we have $\gamma_k = \beta_k = 1$ in Proposition 6, and we recover the average Frobenius error of the randomized SVD (Halko et al., 2011, Thm. 10.5) up to a factor of $(k+p)$ due to the non-independence of $\mathbf{\Omega}_1$ and $\mathbf{\Omega}_2$ in general.

**Lemma 7** *With the notations introduced at the beginning of the section, for all $s \geq 0$, we have*

$$\mathbb{P}\left\{\|\mathbf{\Sigma}_2\mathbf{\Omega}_2\|_{\mathrm{F}}^2 > \ell(1+s)\,\mathrm{Tr}(\mathbf{\Sigma}_2^2\mathbf{V}_2^*\mathbf{KV}_2)\right\} \leq (1+s)^{\ell/2}e^{-s\ell/2}.$$

**Proof.** Let $\omega_j$ be the $j$th column of $\mathbf{\Omega}$ for $1 \leq j \leq \ell$ and $v_1, \ldots, v_n$ be the $n$ columns of the orthonormal matrix $\mathbf{V}$. We first remark that

$$\|\mathbf{\Omega}_2\|_{\mathrm{F}}^2 = \sum_{j=1}^{\ell} Z_j, \qquad Z_j \coloneqq \sum_{n_1=1}^{n-k} \sigma_{k+n_1}^2(\mathbf{A})(v_{k+n_1}^*\omega_j)^2, \tag{9}$$

where the $Z_j$ are i.i.d because $\omega_j \sim \mathcal{N}(0, \mathbf{K})$ are i.i.d. Let $\lambda_1 \geq \cdots \geq \lambda_n \geq 0$ be the eigenvalues of $\mathbf{K}$ with eigenvectors $\psi_1, \ldots, \psi_n \in \mathbb{R}^n$. For $1 \leq j \leq \ell$, we have,

$$\omega_j = \sum_{i=1}^{n}(c_i^{(j)})^2\sqrt{\lambda_i}\psi_i,$$

where $c_i^{(j)} \sim \mathcal{N}(0, 1)$ are i.i.d for $1 \leq i \leq n$ and $1 \leq j \leq \ell$. Then,

$$Z_j = \sum_{i=1}^{n}(c_i^{(j)})^2\lambda_i\sum_{n_1=1}^{n-k}\sigma_{k+n_1}^2(\mathbf{A})(v_{k+n_1}^*\psi_i)^2 = \sum_{i=1}^{n} X_i$$

where the $X_i$ are independent. Let $\gamma_i = \lambda_i \sum_{n_1=1}^{n-k} \sigma_{k+n_1}^2(\mathbf{A})(v_{k+n_1}^* \psi_i)^2$, then $X_i \sim \gamma_i \chi^2$ for $1 \le i \le n$.

Let $0 < \theta < 1/(2 \sum_{i=1}^n \gamma_i)$, we can bound the moment generating function of $\sum_{i=1}^n X_i$ as

$$\mathbb{E}\left[e^{\theta \sum_{i=1}^n X_i}\right] = \prod_{i=1}^n \mathbb{E}\left[e^{\theta X_i}\right] = \prod_{i=1}^n (1 - 2\theta\gamma_i)^{-1/2} \le \left(1 - 2\theta \sum_{i=1}^n \gamma_i\right)^{-1/2}$$

because the $X_i/\gamma_i$ are independent and follow a chi-squared distribution. The right inequality is obtained by showing by recurrence that, if $a_1, \ldots, a_n \ge 0$ are such that $\sum_{i=1}^n a_i \le 1$, then $\prod_{i=1}^n (1 - a_i) \ge 1 - \sum_{i=1}^n a_i$. For convenience, we define $C_1 := \sum_{i=1}^n \gamma_n$, we have shown that

$$\mathbb{E}\left[e^{\theta Z_j}\right] \le (1 - 2\theta C_1)^{-1/2}.$$

Moreover, we find that

$$C_1 = \sum_{n_1=1}^{n-k} \sigma_{k+n_1}^2 v_{k+n_1}^* \left(\sum_{i=1}^n \psi_i^* \lambda_i \psi_i\right) v_{k+n_1} = \sum_{n_1=1}^{n-k} \sigma_{k+n_1}^2(\mathbf{A}) v_{k+n_1}^* \mathbf{K} v_{k+n_1}$$
$$= \mathrm{Tr}(\mathbf{\Sigma}_2^2 \mathbf{V}_2^* \mathbf{K} \mathbf{V}_2).$$

Let $s \ge 0$ and $0 < \theta < 1/(2C_1)$, by Chernoff bound (Chernoff, 1952, Thm. 1), we obtain

$$\mathbb{P}\left\{\|\mathbf{\Sigma}_2 \mathbf{\Omega}_2\|_{\mathrm{F}}^2 > \ell(1+s)C_1\right\} \le e^{-(1+s)C_1 \ell\theta} \mathbb{E}\left[e^{\theta Z_j}\right]^\ell$$
$$= e^{-(1+s)C_1 \ell\theta}(1 - 2\theta C_1)^{-\ell/2}.$$

We minimize the bound over $0 < \theta < 1/(2\,\mathrm{Tr}(K))$ by choosing $\theta = s/(2(1+s)C_1)$, which gives

$$\mathbb{P}\left\{\|\mathbf{\Sigma}_2 \mathbf{\Omega}_2\|_{\mathrm{F}}^2 > \ell(1+s)C_1\right\} \le (1+s)^{\ell/2} e^{-\ell s/2}.$$

∎

The proof of Theorem 2 will require to bound the term $\|\mathbf{\Omega}_1^\dagger\|_{\mathrm{F}}^2$, which we achieve using the following result (Boullé & Townsend, 2022, Lem. 3).

**Lemma 8 (Boullé & Townsend 2022)** *Let $k, \ell \ge 1$ such that $\ell - k \ge 4$, with the notations introduced at the beginning of the section, for all $t \ge 1$, we have*

$$\mathbb{P}\left\{\|\mathbf{\Omega}_1^\dagger\|_{\mathrm{F}}^2 > 3t^2 \frac{\mathrm{Tr}((\mathbf{V}_1^* \mathbf{K} \mathbf{V}_1)^{-1})}{\ell - k + 1}\right\} \le t^{-(\ell-k)}.$$

We now prove Theorem 2, which provides a refined probability bound for the performance of the generalized randomized SVD on matrices.

**Proof of Theorem 2.** Using (Boullé & Townsend, 2022, Thm. 2) and the submultiplicativity of the Frobenius norm, we have

$$\|\mathbf{A} - \mathbf{Q}\mathbf{Q}^*\mathbf{A}\|_{\mathrm{F}}^2 \le \|\mathbf{\Sigma}_2\|_{\mathrm{F}}^2 + \|\mathbf{\Sigma}_2 \mathbf{\Omega}_2\|_{\mathrm{F}}^2 \|\mathbf{\Omega}_1^\dagger\|_{\mathrm{F}}^2. \tag{10}$$

Let $\ell = k + p$ with $p \ge 4$, combining Lemmas 7 and 8 to bound the terms $\|\mathbf{\Sigma}_2 \mathbf{\Omega}_2\|_{\mathrm{F}}^2$ and $\|\mathbf{\Omega}_1^\dagger\|_{\mathrm{F}}^2$ in Equation (10) yields the following probability estimate:

$$\|\mathbf{A} - \mathbf{Q}\mathbf{Q}^*\mathbf{A}\|_{\mathrm{F}}^2 \le \|\mathbf{\Sigma}_2\|_{\mathrm{F}}^2 + 3t^2(1+s)\frac{k+p}{p+1}\,\mathrm{Tr}((\mathbf{V}_1^* \mathbf{K} \mathbf{V}_1)^{-1})\,\mathrm{Tr}(\mathbf{\Sigma}_2^2 \mathbf{V}_2^* \mathbf{K} \mathbf{V}_2)$$
$$\le \left(1 + 3t^2(1+s)\frac{(k+p)k}{p+1}\frac{\beta_k}{\gamma_k}\right) \sum_{j=k+1}^n \sigma_j^2(\mathbf{A}),$$

with failure probability at most $t^{-p} + (1+s)^{(k+p)/2} e^{-s(k+p)/2}$. Note that we introduced $\gamma_k := k/(\lambda_1 \mathrm{Tr}((\mathbf{V}_1^* \mathbf{K} \mathbf{V}_1)^{-1}))$ and $\beta_k := \mathrm{Tr}(\mathbf{\Sigma}_2^2 \mathbf{V}_2^* \mathbf{K} \mathbf{V}_2)/(\lambda_1 \|\mathbf{\Sigma}_2\|_{\mathrm{F}}^2)$. We conclude the proof by defining $u = \sqrt{1+s} \ge 1$. ∎

The following Lemma provides an estimate of the quantity $\beta_k$ introduced in the statement of Theorem 2.

**Lemma 9** *Let $\beta_k = \mathrm{Tr}(\mathbf{\Sigma}_2^2 \mathbf{V}_2^* \mathbf{K} \mathbf{V}_2)/(\lambda_1 \|\mathbf{\Sigma}_2\|_{\mathrm{F}}^2)$, then the following inequality holds*

$$\beta_k \leq \sum_{j=k+1}^{n} \frac{\lambda_{j-k}}{\lambda_1} \sigma_j^2(\mathbf{A}) \bigg/ \sum_{j=k+1}^{n} \sigma_j^2(\mathbf{A}).$$

**Proof.** Let $\mu_1 \geq \cdots \geq \mu_{n-k}$ be the eigenvalues of the matrix $\mathbf{V}_2^* \mathbf{K} \mathbf{V}_2$. Using Von Neumann's trace inequality (Mirsky, 1975; Von Neumann, 1937), we have

$$\mathrm{Tr}(\mathbf{\Sigma}_2^2 \mathbf{V}_2^* \mathbf{K} \mathbf{V}_2) \leq \sum_{j=k+1}^{n} \mu_{j-k} \sigma_j^2(\mathbf{A}).$$

Then, the matrix $\mathbf{V}_2^* \mathbf{K} \mathbf{V}_2$ is a principal submatrix of $\mathbf{V}^* \mathbf{K} \mathbf{V}$, which has the same eigenvalues of $K$. Therefore, by (Kato, 2013, Thm. 6.46), the eigenvalues of $\mathbf{V}_2^* \mathbf{K} \mathbf{V}_2$ are individually bounded by the eigenvalues of $\mathbf{K}$, i.e., $\mu_j \leq \lambda_j$ for $1 \leq j \leq n - k$, which concludes the proof. ∎

Finally, we highlight that the statement of Theorem 2 can be simplified by choosing $p = 5$, $t = 4$, and $u = 3$.

**Corollary 1 (Generalized randomized SVD)** *Let $\mathbf{A}$ be an $n \times n$ matrix and $k \geq 1$ an integer. If $\mathbf{\Omega} \in \mathbb{R}^{n \times (k+5)}$ is a Gaussian random matrix, where each column is sampled from a multivariate Gaussian distribution with symmetric positive semi-definite covariance matrix $\mathbf{K} \in \mathbb{R}^{n \times n}$, and $\mathbf{QR} = \mathbf{A\Omega}$ is the economized QR decomposition of $\mathbf{A\Omega}$, then*

$$\mathbb{P}\left[\|\mathbf{A} - \mathbf{QQ}^* \mathbf{A}\|_{\mathrm{F}} \leq \left(1 + 9\sqrt{k(k+5)\frac{\beta_k}{\gamma_k}}\right)\sqrt{\sum_{j=k+1}^{n} \sigma_j^2(\mathbf{A})}\right] \geq 0.999.$$

In contrast, a simplification of Theorem 1 by choosing $t = 6$ and $u = 4$ gives the following result.

**Corollary 2 (Randomized SVD)** *Let $\mathbf{A}$ be an $n \times n$ matrix and $k \geq 1$ an integer. If $\mathbf{\Omega} \in \mathbb{R}^{n \times (k+5)}$ is a standard Gaussian random matrix and $\mathbf{QR} = \mathbf{A\Omega}$ is the economized QR decomposition of $\mathbf{A\Omega}$, then*

$$\mathbb{P}\left[\|\mathbf{A} - \mathbf{QQ}^* \mathbf{A}\|_{\mathrm{F}} \leq \left(1 + 16\sqrt{k+5}\right)\sqrt{\sum_{j=k+1}^{n} \sigma_j^2(\mathbf{A})}\right] \geq 0.999.$$

## B  CONTINUITY OF THE JACOBI KERNEL

Here, we show that if the kernel's eigenvalue sequence, $\{\lambda_j\}$, is such that $\sum_{j=1}^{\infty} j\lambda_j < \infty$, then the Jacobi kernel $K_{\mathrm{Jac}}^{(2,2)}$ defined by Equation (5) is continuous. First, note that $\tilde{P}_j^{(2,2)}$ is a scaled ultraspherical polynomial $\tilde{C}_j^{(5/2)}$ with parameter $5/2$ and degree $j \geq 0$ so it can be bounded by the following proposition.

**Proposition 10** *Let $\tilde{C}_j^{(5/2)}$ be the ultraspherical polynomial of degree $j$ with parameter $5/2$, normalized such that $\int_{-1}^{1}(1 - x^2)^2 \tilde{C}_j^{(5/2)}(x)^2 \, \mathrm{d}x = 1$. Then,*

$$\max_{x \in [-1,1]} |(1 - x^2)\tilde{C}_j^{(5/2)}(x)| \leq 2\sqrt{j + 5/12}, \quad j \geq 0. \tag{11}$$

**Proof.** Let $j \geq 0$ and $x \in [-1, 1]$, according to (NIS, Table 18.3.1),

$$\tilde{C}_j^{(5/2)}(x) = 3\sqrt{\frac{j + 5/2}{(j+1)(j+2)(j+3)(j+4)}} C_j^{(5/2)}(x), \tag{12}$$

where $C_j^{(5/2)}(x)$ is the standard ultraspherical polynomial. Using (NIS, (18.9.8)), we have

$$(1 - x^2)C_j^{(5/2)}(x) = \frac{(j+3)(j+4)C_j^{(3/2)}(x) - (j+1)(j+2)C_{j+2}^{(3/2)}(x)}{6(j + 5/2)}.$$

By using (NIS, (18.9.7)), we have $(C_{j+2}^{(3/2)}(x) - C_j^{(3/2)}(x))/2 = (j + 5/2)C_{j+2}^{(3/2)}(x)$ and hence,

$$(1 - x^2)C_j^{(5/2)}(x) = \frac{2}{3}C_j^{(3/2)}(x) - \frac{(j+1)(j+2)}{3}C_{j+2}^{(1/2)}(x).$$

We bound the two terms with (NIS, (18.14.4)) to obtain the following inequalities:

$$|C_j^{(3/2)}(x)| \leq \frac{(j+1)(j+2)}{2}, \qquad |C_{j+2}^{(1/2)}(x)| \leq 1.$$

Hence, $|(1 - x^2)C_j^{(5/2)}(x)| \leq 2(j+1)(j+2)/3$ and following Equation (12) we obtain

$$|(1 - x^2)\tilde{C}_j^{(5/2)}(x)| \leq 2\sqrt{\frac{(j+1)(j+2)(j+5/2)}{(j+3)(j+4)}} \leq 2\sqrt{j+5/12},$$

which concludes the proof. ■

The bound given in Proposition 10 differs by a factor of $4/3$ from the numerically observed upper bound $(1.5\sqrt{j+5/12})$. This result shows that if $\sum_{j=1}^{\infty} j\lambda_j < \infty$, then the series of functions in Equation (5) converges uniformly and $K_{\text{Jac}}^{(2,2)}$ is continuous.

**Proof of Lemma 3.** By Markov brothers' inequality (Markov, 1889), for all $j \geq 0$ and $0 \leq \mu \leq j$, we have $\max_{x \in [-1,1]} |p_j^{(\mu)}(x)| \leq j^{2\mu}$. Therefore, $|f_n^{(\mu)}(x)| \leq \sum_{j=0}^{n} |a_j| \|p_j^{(\mu)}\|_\infty \leq \sum_{j=0}^{n} j^{2\mu-\nu}$ so $|f_n^{(\mu)}(x)| < \infty$ if $\mu < (\nu - 1)/2$. The result follows from (Rudin, 1976, Thm. 7.17). ■

**Proof of Theorem 4.** Since $f \sim \mathcal{GP}(0, K_{\text{Jac}}^{(2,2)})$, $f \sim \sum_{j=0}^{\infty} c_j \sqrt{\lambda_{j+1}}(1 - x^2)\tilde{P}_j^{(2,2)}(x)$, where $c_j \sim \mathcal{N}(0,1)$ for $j \geq 0$. Let $f_n$ denote the truncation of $f$ after $n$ terms. By letting $M > 0$ be the constant such that $\lambda_{j+1} \leq M(j+1)^{-\nu}$, we find that

$$\|f - f_n\|_\infty \leq S_n, \qquad S_n := 2\sqrt{M} \sum_{j=n+2}^{\infty} |c_{j-1}| j^{(1-\nu)/2},$$

where we used $\max_{x \in [-1,1]} |(1 - x^2)\tilde{P}_j^{(2,2)}(x)| \leq 2\sqrt{j+1}$ (cf. Proposition 10). Thus, we have

$$\mathbb{P}\left(\lim_{n \to \infty} \|f - f_n\|_\infty = 0\right) \geq \mathbb{P}\left(\lim_{n \to \infty} S_n = 0\right).$$

Here, $S_n \sim X_n = \sum_{j=n+2}^{\infty} Y_j j^{(1-\nu)/2}$, where $Y_j$ follows a half-normal distribution (Leone et al., 1961) with parameter $\sigma = 1$ and the $(Y_j)_j$ are independent. We want to show that $X_n \xrightarrow{a.s.} 0$. For $\epsilon > 0$, using Chebyshev's inequality, we have:

$$\sum_{n=0}^{\infty} \mathbb{P}(|X_n| \geq \epsilon) \leq \frac{1}{\epsilon^2} \sum_{n=0}^{\infty} \left(1 - \frac{2}{\pi}\right) \sum_{j=n+2}^{\infty} \frac{1}{j^{\nu-1}} \leq \frac{1}{\epsilon^2}\left(1 - \frac{2}{\pi}\right) \frac{1}{\nu - 2} \sum_{n=1}^{\infty} \frac{1}{n^{\nu-2}},$$

which is finite if $\nu > 3$. Therefore, using Borel–Cantelli Lemma (Durrett, 2019, Chapt. 2.3), $X_n$ converges to 0 almost surely and $\mathbb{P}(\lim_{n \to \infty} X_n = 0) = 1$. Finally,

$$\mathbb{P}\left(\lim_{n \to \infty} \|f - f_n\|_\infty = 0\right) \geq \mathbb{P}\left(\lim_{n \to \infty} X_n = 0\right) = 1,$$

which proves that $\{f_n\}$ converges uniformly and hence $f$ is continuous with probability one. The statement for higher order derivatives follows the proof of Lemma 3. ■

## C  EXPERIMENTS WITH NON-STANDARD COVARIANCE MATRICES

In this section, we provide more detailed explanations regarding the numerical experiments conducted in Section 5.1.

Sampling a random vector from a multivariate normal distribution with an arbitrary covariance matrix $\mathbf{K}$ can be computationally expensive when the dimension, $n$, of the matrix is large as it requires

the computation of a Cholesky factorization, which can be done in $\mathcal{O}(n^3)$ operations. We highlight that this step can be precomputed once, such that the overhead of the generalized SVD can be essentially expressed as the cost of an extra matrix-vector multiplication. Then, the difference in timings between standard and prior covariance matrices is marginal as shown by the right panel of Figure 2. Additionally, we would like to highlight that prior covariance matrices can be designed and derived using physical knowledge of the problem, such as a diffusion behaviour of the system, which can also significantly decrease the precomputation cost. As an example, in Section 5.1, we employ the discretized Green's function of the negative Laplacian operator with homogeneous Dirichlet boundary conditions, given by $\mathcal{L}u = -d^2u/dx^2$ on $[0, 1]$, for which we know the eigenvalue decomposition. Hence, the eigenvalues and normalized eigenfunctions are respectively given by

$$\lambda_n = \frac{1}{\pi^2 n^2}, \qquad \psi_n(x) = \sqrt{2}\sin(n\pi x), \qquad x \in [0, 1], \, n \geq 1.$$

Therefore, one can employ Mercer's representation (see Equation (4)) to sample the random vectors and precompute the covariance matrix in $\mathcal{O}(n^2)$ operations. For a problem of size $n = 2000$, it takes 0.16s to precompute the matrix.

