# OpenReview forum: "A generalization of the randomized singular value decomposition"
_ICLR.cc/2022/Conference — ICLR 2022 Poster_

### Official Review · Reviewer_FGbe · 2021-10-29

**Correctness:** 3
**Technical Novelty And Significance:** 3
**Empirical Novelty And Significance:** 2
**Recommendation:** 5
**Confidence:** 3

**Main Review:**

Basically, I like the paper's motivation; it is quite a fundamental research problem to reduce computational time by using randomized SVD and improve its approximation quality. Besides, this paper is well structured, and the theoretical background of the proposed approach is well described in the paper. However, I have the following several concerns to the paper:

In the experiments, it uses only synthetic datasets. Therefore, it is unclear whether the proposed approach is useful for real applications. Typically, if the given matrix is dense, singular values decay rapidly. If the given matrix is sparse, singular values decay slowly. I want to know whether the proposed approach can achieve high approximation accuracy for various types of matrices.

In the experiment of Figure 2, it seems that the computational time of the Cholesky factorization is not included. I think this is not a fair comparison to the previous approach. Since the computational cost of the Cholesky factorization is cubic according to the size of the covariance matrix, the proposed approach would need a large computational time for the Cholesky factorization. In contrast, the previous approach does not need to compute Cholesky factorization. I want to know the end-to-end computational times of the proposed and previous approach for the specific target rank of SVD.
In terms of computational time, the paper should show the theoretical computational cost of the proposed approach.

As for the approximation quality, the previous approach, as well as the proposed approach, can more accurately perform SVD as the oversampling parameter increases. So, I think the paper should show the experimental results of the approximation quality of each approach against the oversampling parameter. The setting of the oversampling parameter in the experiments is unclear from the descriptions of the paper.

**Summary Of The Paper:**

This paper proposes a generalization approach for the randomized SVD. In the existing approach of the randomized SVD, a standard Gaussian random matrix is used to reduce the size of a matrix to perform SVD. The proposed approach uses a multivariate Gaussian distribution with a covariance matrix instead of the standard Gaussian random matrix in performing SVD. Since the covariance matrix can reflect prior knowledge of the given matrix, the proposed approach can improve the approximation quality of the randomized SVD. By using a synthetic dataset, this paper conducts experiments to show the effectiveness of the proposed approach.

**Summary Of The Review:**

This paper is well-written.
Real datasets should be used in the experiments.
The computational time of the Cholesky factorization should be included in Figure 2.

---

> ### Author Response · Authors · 2021-11-17
> **Response to Reviewer FGbe [1/2]**
>
> We thank the reviewer for the detailed review and the positive comments regarding the motivation and structure of the paper. We address the concerns of the reviewer below.
>
> > In the experiments, it uses only synthetic datasets. Therefore, it is unclear whether the proposed approach is helpful for real applications. Typically, if the given matrix is dense, singular values decay rapidly. If the given matrix is sparse, singular values decay slowly. I want to know whether the proposed approach can achieve high approximation accuracy for various types of matrices.
>
> The reviewer is correct to write that the singular values decay rate of the matrix might depend on the application considered. However, this is not a limitation of our method but rather an intrinsic fact about low-rank approximation: the best approximation error of a matrix by a low-rank matrix is given by the tail of the singular values (Eckart-Young theorem). Hence, one cannot achieve high accuracy to approximate a matrix with slowly decaying singular values with a small number of matrix-vector products. This limitation is already present in the original randomized SVD paper by Halko-Martinsson-Tropp, which has been widely applied over the past decade in various fields such as biology [1], imaging [2], machine learning [3], fluid mechanics [4],... (the paper has been cited over 3000 times).
>
> In addition, as the Reviewer ksBY wrote, the rank of large matrices is often small, i.e., singular values decay fast. This fact is also explained in the recent paper by Udell & Townsend, which shows that a low-rank matrix can often approximate large matrices. Then, the Eckart-Young theorem implies a fast decay rate of the singular values and shows that our approach, more generally the randomized SVD, is applicable in this setting.
>
> [1] Hie, Bryson, Berger, Efficient integration of heterogeneous single-cell transcriptomes using Scanorama, Nature Biotechnology, 2019.
>
> [2] Li et al., Machine-learning reprogrammable metasurface imager, Nature Communications, 2019.
>
> [3] Maddox et al., A Simple Baseline for Bayesian Uncertainty in Deep Learning, NeurIPS 2019.
>
> [4] Brunton et al., Machine Learning for Fluid Mechanics, Annual Review of Fluid Mechanics, 2020.

---

> > ### Author Response · Authors · 2021-11-17
> > **Response to Reviewer FGbe [2/2]**
> >
> > > In the experiment of Figure 2, it seems that the computational time of the Cholesky factorization is not included. I think this is not a fair comparison to the previous approach. Since the computational cost of the Cholesky factorization is cubic according to the size of the covariance matrix, the proposed approach would need a large computational time for the Cholesky factorization. In contrast, the previous approach does not need to compute Cholesky factorization. I want to know the end-to-end computational times of the proposed and previous approach for the specific target rank of SVD. In terms of computational time, the paper should show the theoretical computational cost of the proposed approach.
> >
> > This is a great comment raised by the reviewer. As we write in the paper, we are primarily interested in this experiment in applications where the computational time needed to perform the randomized SVD is much smaller than the time required to sample the matrix. For example, for practical experiments where it's costly to acquire new data, one would get the best possible accuracy from the fixed number of samples available). Moreover, we highlight that the prior kernel, in this case, is the Green's function of the Laplacian operator, from which we already know the eigenvalues and eigenvectors:
> > $$\lambda_n=\frac{1}{n^2},\qquad \psi_n(x)=\sqrt{2}\sin(n\pi x),\qquad n\geq , x\in[0,1].$$
> > Then, one can use Mercer's representation to sample random vectors from the corresponding normal distribution.
> >
> > This choice of the kernel is convenient for two reasons: first, it imposes a prior on a system with diffusion behavior, and secondly, it's extremely fast to sample the associated Gaussian process using Mercer's representation. Then, the overhead in the computational time is essentially a matrix-vector product. Moreover, this kernel is very natural and already employed in many deep learning techniques that learn solution operators of PDEs [1,2,3].
> >
> > In the updated version of the manuscript, we recomputed Fig. 2 using the known eigenvalue decomposition of the Green's function. The slight increase of the error towards many samples is due to the discretization error of the matrix $A$ that we wish to learn. This is not significant for the following reasons: (1) in practical applications, we often want to learn a rank $k$ representation of the $n\times n$ matrix $A$ where $k<<n$, (2) we do not want to learn the discretization error of the matrix $A$. We also added a paragraph in Appendix C to explain how we compute the covariance matrix and give the corresponding timings as asked by the reviewer.
> >
> > [1] Li et al., Fourier neural operator for parametric partial differential equations, ICLR 2021.
> >
> > [2] Gin et al. DeepGreen: deep learning of Green’s functions for nonlinear boundary value problems, Scientific Reports, 2021.
> >
> > [3] Kovachi et al. Neural Operator: Learning Maps Between Function Spaces, arXiv:2108.08481, 2021.
> >
> > > As for the approximation quality, the previous approach, as well as the proposed approach, can more accurately perform SVD as the oversampling parameter increases. So, I think the paper should show the experimental results of the approximation quality of each approach against the oversampling parameter. The setting of the oversampling parameter in the experiments is unclear from the descriptions of the paper.
> >
> > We thank the reviewer for the comment and highlight that the distinction between the rank $k$ and the oversampling parameter $p$ is not relevant for practical use since one cannot control the target rank $k$. Then, what matters for practical experiments is the total number of samples $k+p$, i.e., the number of columns of the Gaussian matrix $\Omega$, that one uses to approximate the matrix.

---

### Official Review · Reviewer_ksBY · 2021-11-03

**Correctness:** 4
**Technical Novelty And Significance:** 4
**Empirical Novelty And Significance:** 3
**Recommendation:** 8
**Confidence:** 4

**Main Review:**

Randomized Singular Value Decomposition (SVD) is an important algorithm in the field of numerical linear algebra. Suppose $A \in \mathbb{R}^{m \times n}$ is a large matrix. It is often the case that although $m$ and $n$ are large, the rank of the matrix $A$ is small, and therefore a low rank approximation $A \approx B C$ for $B \in \mathbb{R}^{m \times k}$ and $A \in \mathbb{R}^{k \times n}$ and $k \ll \min(m,n)$ is desired. Randomized SVD computes a rank-$k$ approximation for $A$. The recipe for doing that is to first compute an approximate basis for the range space of $A$. Specifically, we want an orthonormal matrix $Q \in \mathbb{R}^{m \times k}$ such that $A \approx Q Q^* A$. Randomized SVD does this by letting $\Omega \in \mathbb{R}^{n \times k}$ be a matrix whose entries are i.i.d. standard Gaussian, and then letting $Q$ be the orthonormal matrix from the QR-decomposition $Q R = A \Omega$. Once we have $Q$, we have a low rank approximation $A \approx Q B$ for $B = Q^* A$. The next step is to compute the SVD of the small matrix $B = \tilde{U} \Sigma V*$, which finally gives the approximate SVD for $A$ as $U \Sigma V^*$ for $U = Q \tilde{U}$.

This paper starts by first generalizing the above recipe by letting the columns of $\Omega$ to be sampled from a multivariate Gaussian distribution with a general covariance matrix $K \in \mathbb{R}^{n \times n}$. The error $\lVert A - Q Q^* A \rVert_F$ is upper bounded, which is similar to the upper bound in vanilla Randomized SVD. Then this method is also generalized to approximation of Hilbert-Schmidt operators $\mathscr{F}$, instead of matrices $A$. The samples are now coming from a Gaussian process with a covariance kernel $K$. As is expected, the approximation quality of the randomized algorithms is closely tied to the spectrum of the covariance matrix/kernel $K$. To this end, some intuition is provided for the relation between the choice of the covariance kernel and the approximation ability of the algorithm. Finally, numerical experiments provide justification for these generalizations by showcasing their power.

The only minor complaint about the paper is that the presentation could be made less terse, especially in Section 3.1 where Randomized SVD for Hilbert-Schmidt operators is discussed. For example, it would be nice to provide some background on how QR algorithm is used on $Y$ to get $Q$, or why each column of $\Omega$ consists of "polynomial representation of a smooth random function sampled from the GP in the Chebyshev basis". It's also unfortunate that no results are provided for the approximation errors in the case of Hilbert-Schmidt operators.

I also didn't understand why the quantity $\gamma_k$ measures "the quality of the covariance kernel $K$ in $\mathcal{GP}(0, K)$ to generate random functions that can learn the H-S operator $\mathscr{F}$" claimed in Section 4.2.

**Summary Of The Paper:**

This paper proposes and analyzes a generalization of the randomized SVD to incorporate any covariance matrix and to Hilbert-Schmidt operators. Extensive numerical experiments further strengthen the case for this generalization.

**Summary Of The Review:**

The paper should be accepted because it is well-written for the most part, tackles an important problem, and provides a novel generalization of a well-known method.

---

> ### Author Response · Authors · 2021-11-17
> **Response to Reviewer ksBY**
>
> > The only minor complaint about the paper is that the presentation could be made less terse, especially in Section 3.1 where Randomized SVD for Hilbert-Schmidt operators is discussed. For example, it would be nice to provide some background on how QR algorithm is used on  to get , or why each column of  consists of "polynomial representation of a smooth random function sampled from the GP in the Chebyshev basis".
>
> We thank the reviewer for the very positive comments about our paper. We agree with the referee about the presentation in Section 3.1, but unfortunately, the space constraint limits us, and we have to focus on the main points. We added a reference to the continuous QR algorithm used in Chebfun (Townsend & Trefethen, 2015) and explained how we use it in Algorithm 1 of the paper. Concerning the columns of $\Omega$, Chebfun approximates smooth functions by polynomials up to machine precision to apply continuous analogues of matrix operations. Then, the columns on $\Omega$ consist of a polynomial approximation of random functions sampled from the GP. We replaced the word "representation" with "approximation" to clarify this point.
>
> > It's also unfortunate that no results are provided for the approximation errors in the case of Hilbert-Schmidt operators.
>
> We highlight in point 1. of the Contributions and the beginning of Section 3 that the probability bounds proved in this paper also hold for Hilbert-Schmidt operators, giving theoretical guarantees for the HS randomized SVD presented in Section 3.1. We chose to only state the results once for matrices as the HS results are identical. Moreover, as the reviewer wrote, the randomized SVD is an essential algorithm in numerical linear algebra. Its generalization and new theoretical bounds may find direct applications for approximating large matrices efficiently by incorporating prior knowledge on the design of the covariance matrix.
>
> > I also didn't understand why the quantity  measures "the quality of the covariance kernel  in  to generate random functions that can learn the H-S operator " claimed in Section 4.2.
>
> Thanks for the comment. The quantity $\gamma_k$ measures the ability of the covariance kernel to capture the first $k$ right singular vectors of $A$, $V_1$. Then, it measures how close the eigenfunctions of $K$ are to $V_1$. In the ideal case, where we know $V_1$, then we can select the first $k$ eigenfunctions of $K$ to be equal to the first $k$ right singular functions of $A$ and have $\lambda_1=...=\lambda_k=1$ so that $\gamma_k = 1$. Of course this very specific case will not necessarily happen in practice. Similarly, the other quantity $\beta_k$ measures the relation between the last right singular vectors of $A$, $V_2$, and the eigenfunctions of $K$. This shows that imposing prior knowledge of $A$ through the covariance matrix can decrease the approximation error, as shown by our experiments in section 5.1. In the continuous setting, where the aim is to approximate a kernel of a HS operator, prior information might be enforced by choosing a kernel whose eigenfunctions are polynomials of increasing degree, which is what we do with the Jacobi kernel.

---

> > ### Comment · Reviewer_ksBY · 2021-11-28
> > **Response to authors**
> >
> > Thank you for your response. It answers my questions.

---

### Official Review · Reviewer_uuEP · 2021-11-03

**Correctness:** 3
**Technical Novelty And Significance:** 2
**Empirical Novelty And Significance:** 3
**Recommendation:** 8
**Confidence:** 3

**Main Review:**


0) In abstract: "Here, we generalize the theory of randomized SVD to to multivariate Gaussian vectors, [...]"

- A theory of randomized SVD already exists for multivariate Gaussian vectors though? Furthermore, read 1).

1) "Generalization 1 [...]"

- This bit is unclear. It seems like the authors are claiming to generalise the randomised SVD for Gaussian vectors with non-identity covariance. However, in the beginning of Sec 3 this was already developed by Boulle & Townsend 2021. I guess what is meant is that new theoretical results pertaining to the approximation error are obtained? I would expect this to be more clear in a  camera-ready version. Also I guess this is the same issue as in abstract (point 0) above).

2) "In particular, this highlights that a suitably chosen covariance matrix can outperform the randomized SVD with standard Gaussian vectors"

- Does the converse hold as well? i.e. can an ill-suited covariance matrix make the algorithm worse?

3) "Therefore, each column of \Omega is an object, consisting of a polynomial representation [...]"

- Why polynomials instead of something else?

4) Bad figure ordering / Layout.

-The first Figure to be references is Fig. 4 (rather than 1) on page 4. However, the actual figure does not appear until page 9.

5) Bad notation.

-L is used both to denote the Cholesky factor of a kernel and the space of square integrable functions. E.g. L^2(D) could be interpreted as the square of the Cholesky factor evaluated at D.

6) Bad enumeration notation. In the first paragraph of page 5 items are enuemrated as (1), (2), and (3). However this is the same notation used to reference numbered equations making the paragraph quite hard to read.

7) "A desireable property of a covariance kernel is to be unbiased towards one spatial direction"

-This would require further motivation. Would it not depend on application / prior knowledge of the problem?

8) "[...], we observe a large variation of the randomly generated functions near x = \pm 1, [...]"

-Upon examining Figure 1 it seems to me the opposite is true, i.e. the largest variation is found in mid-interval while it is the smallest at the end-points. Please clarify.

9) Figure 2.

- I believe the authors might get their point across better if (additionally) the error is plotted vs time.

10) Proofs.

- The proofs would be easier to follow if the authors actually state what results they are using in their arguments instead of e.g. just referencing (Halko et al, 2011 Prop A.1).

11) Proof of Lemma 7.

-It is not stated what is used how to retrieve the upper bound of the moment generating function.

12) Proof of Theorem 2. "[...] combining (Boulle & Townsend, 2021, Thm 3.2) and Lemma 7 in Equation (10) [...]"

- Lemma 7 is not given in Equation (10). Also the proof would be more readable if the result by Boulle and Townsend was stated in the text.

13)



**Summary Of The Paper:**

This paper studies the randomised SVD algorithm with non-standard Gaussian vectors which is then used to approximate Hilbert-Schmidt operators using a new kernel based on out products of weighted Jacobi polynomials.

**Summary Of The Review:**

While the paper appears to adduce some novel results. However, the presentation is unclear which makes it hard to verify the results of the authors and properly situate it in the context of previous literature.

I do however think that the authors have a reasonable chance to address my concerns for a camera-ready version, why I recommend a marginal accept.

---

> ### Author Response · Authors · 2021-11-17
> **Response to Reviewer uuEP**
>
>
> We thank the reviewer for the positive feedback and detailed review. We address each of the individual points raised below and updated the paper accordingly.
>
> 0. We updated the abstract to state that "we generalized the randomized SVD." Here, we mainly refer to the algorithm and numerical results, which have not been considered by Boulle & Townsend and the new theoretical bounds.
> 1. We are sorry for the confusion and removed the word "theory". Similarly to point 0, we generalize the algorithm to work with multivariate Gaussian inputs, observe improved accuracy, and provide new probability bounds for the randomized SVD. This shows that one can exploit prior knowledge on the matrix to obtain lower approximation error. This observation would not be possible with the prior bounds developed by Boulle & Townsend, and is the primary motivation for using non-standard Gaussian vectors in the matrix case. Additionally, we develop an algorithm for Hilbert-Schmidt operators and provide numerical results. Our contributions are stated clearly at the end of the introduction.
> 2. Yes, absolutely. This is an excellent comment; indeed, the performance of the randomized SVD can be made arbitrarily bad if the covariance matrix is not chosen carefully. As an example, take the matrix $A=U\Sigma V^*$, if the covariance matrix has all its eigenvectors orthogonal to $v_1$: the first right singular vector of $A$. Then one cannot hope to capture the first singular values of $A$ using the randomized SVD. One of the paper's messages is that the choice of the covariance matrix matters for applications: a suitably chosen one can lead to significant improvement, while the error can be made arbitrarily large if one is not careful.
> 3. The columns of $\Omega$ do not have to be polynomials. However, we implemented the algorithm in Chebfun due to its capabilities for computing with smooth functions. Chebfun approximates smooth functions by Chebyshev polynomials to matrix precision to use fast algorithms for working with polynomials. Then, when selecting the squared-exponential covariance kernel, Chebfun is constructing a bivariate polynomial approximation of the kernel, accurate to 16 digits.
> 4. We agree with the reviewer and therefore remove the reference to Fig 4 in p.4, which was used for illustration purposes only. We prefer to maintain Fig.4 in section 5.2 of the paper to
> 5. We apologize for the confusion. We changed the notation for the continuous analogue of the Cholesky factor in the revised version to $L_c$.
> 6. We agree with the reviewer and changed the enumeration notation to "1., 2., 3."
> 7. This indeed depends on the application or prior knowledge of the problem. In this case, we were thinking of a general case where we have little prior knowledge of the problem and do not know whether the operator has a specific and different behavior on the positive or negative real line, which is why we impose symmetry of the covariance kernel. Of course, one could employ a Jacobi kernel with $\alpha\neq \beta$ in more specific cases.
> 8. By large variations, we mean maxima of the derivative: in the bottom right plot of Fig.1, one can see on the purple curve that the derivative becomes extremely large near $x=1$. This is because the random function is not guaranteed to be continuous around $x=1$. Note that we truncated the series of functions to $n=500$, but as $n\to\infty$, we should observe a discontinuity at $x=1$: $f(1)\neq \lim_{x\to 1}f(x)$. On the contrary, functions sampled with a kernel with faster eigenvalue decay have smoothness guarantees, as shown in the bottom left plot of Fig.1.
> 9. Thanks for the suggestion. We made the Figure, but the output was quite confusing as each point corresponds to a different problem since the number of samples varies. Therefore, we decided not to include it in the paper to avoid confusion for the reader. We provide the Figure in the Supplementary Material for the reviewer. We are willing to have it in the camera-ready version of the paper if the reviewer feels that it is necessary.
> 10. We added more explanations in the proofs, as suggested. We have also been informed that the expectation bound of Thm 1.1 in Boulle & Townsend contains a small typo affecting the constant, so we updated Prop. 6 accordingly. The central probability results remain unchanged.
> 11. We added a sentence in the proof to explain how we obtain the upper bound of the moment-generating function.
> 12. Sorry for the confusion, we meant that we bound the two terms $||\Sigma_2\Omega_2||_F^2$ and $||\Omega_1^\dagger||_F^2$ from Eq. (10) using both Lem. 3.5 from Boulle and Townsend and Lemma 7. We updated the sentence to clarify this point. In addition, we now state Lem 3.5 from Boulle and Townsend in the Appendix to make the proof easier to read.

---

> > ### Comment · Reviewer_uuEP · 2021-11-29
> >
> > I would like to thank the authors for their clarifications. Looks like its going to be a good paper in the end.

---

### Decision · Program_Chairs · 2022-01-20

**Decision:**

Accept (Poster)

**Comment:**

This paper studies a generalization of the randomized SVD algorithm with non-standard Gaussian vectors, which is then used to incorporate any covariance matrix and to Hilbert-Schmidt operators.  It uses a new kernel related to products of weighted Jacobi polynomials; and extensive numerical experiments further strengthen the case for this generalization.  Reviewers had initial concerns that were addressed, and the method should be of broad interest.